# Determination of Nutritional and Antioxidant Properties of Maya Nut Flour (*Brosimum alicastrum*) for Development of Functional Foods

**DOI:** 10.3390/foods12071398

**Published:** 2023-03-25

**Authors:** Carolina Losoya-Sifuentes, Karen Pinto-Jimenez, Mario Cruz, Rosa M. Rodriguez-Jasso, Hector A. Ruiz, Araceli Loredo-Treviño, Claudia Magdalena López-Badillo, Ruth Belmares

**Affiliations:** 1Functional Foods & Nutrition Research Group, Food Research Department, School of Chemistry, Universidad Autónoma de Coahuila, Boulevard Venustiano Carranza and José Cárdenas s/n, República Oriente, Saltillo 25280, CP, Mexico; 2Department of Food Science and Technology, Universidad Autónoma Agraria Antonio Narro, Calzada Antonio Narro 1923, Colonia Buenavista, Saltillo 25315, CP, Mexico

**Keywords:** drying methods, Maya nut flour, physicochemical properties, antioxidant activity

## Abstract

Maya nut (*Brosimum alicastrum*) is a novel food with high nutritional value. This research aimed to evaluate the nutritional and antioxidant properties of Maya nut flour (MNF) made from seeds dried by different methods (sun-dried and using hot air at 45 °C and 60 °C) to explore its incorporation into cookies and evaluate its nutritional and functional properties. The naturally sun-dried flour (NF) had the highest content of ash (3.64 ± 0.11 g/100 g), protein (6.35 ± 0.44 g/100 g), crude fiber (6.75 ± 0.29 g/100 g), and functional properties (water and oil absorption). The color of the flour was affected by the different drying methods. While the drying methods influenced the total polyphenolic content (TPC) and antioxidant activity (AA) of MNF, they did not affect the morphology of the native starch or generated important molecular-structural changes. The substitution of 60% of wheat flour with NF in the cookie’s formula increased the protein and fiber content, whereas 20% substitution increased its AA. MNF is a source of protein, dietary fiber, micronutrients, and functional compounds that can enrich cookie formulations.

## 1. Introduction

In recent years, new functional foods or ingredients have become popular due to the increasing awareness of their positive health effects. One of these foods is the Ramon seed (*Brosimun alicastrum*). According to historical data, the Mayas cultivated the Ramon tree to use its seeds as food in their diet. Currently, the Ramon tree is an underexploited resource with high economic potential. In Mexico, conserving and restoring the Ramon tree is sought in forest systems. Its wood, leaves, fruits, and latex are used as a source of food and in traditional medicine [1].

The Ramon tree produces a sweet-tasting fruit with a single seed, known as Maya nut [2]. These seeds (cooked or roasted) are eaten alone, mixed with maize flour, or sweetened with honey to prepare various foods and hot drinks [3]. The nutritional composition of the Maya nut comprises 10.49% fat, 5.21% crude fiber, and 2.02% crude protein in addition to minerals such as copper, potassium, iron, zinc, and sodium [4,5].

Several phenolic compounds, such as gallic acid, p-hydroxybenzoic acid, vanillic acid, caffeic acid, and p-coumaric acid, have been determined to have antioxidant activity [1,2,3,4,5].

Previous research findings indicate that the Maya nut has higher total phenolic content (TPC) and antioxidant activities (AA) than TPC and AA of commercial nuts such as walnut, almond, and peanut [5]; therefore, Maya nut can also be considered a functional ingredient [6]. Recently, consumption of antioxidant-rich food has increased due to their fundamental role in the prevention of various diseases [7].

Flour is the main product from processed Maya nut. As flour, it can be stored for long periods and be used in the manufacture of different food products [8]. Drying is described as a process of removing moisture; reducing the moisture in food inhibits microbial growth and reduces biochemical reactions that could spoil it [9]. However, this process leads to physicochemical changes that influence the color, flavor, texture, and overall quality of the dried product [10]. Different drying methods, such as sun drying, hot air drying, and oven drying, have been used in different plant materials, and their effects on the techno-functional, nutritional, and phytochemical characteristics have been evaluated [9,11,12,13]. To date, there are no data about the effects of different drying methods and conditions on the physical, nutritional, and antioxidant characteristics of the Maya nut.

Currently, consumers demand foodstuffs with functional and nutraceutical properties, and not foodstuffs that only meet the minimum nutritional requirements [10]. Cookies are a highly consumed food, but they are often made with wheat flour (whole wheat or refined) which has very little dietary fiber and, since it is deficient in the essential amino acid lysine, lacks good quality protein. Nevertheless, cookies may be the right product to reformulate to promote health and meet dietary demands. Cookies can be redesigned to utilized a natural product that enriches its nutritional quality and functional properties. Therefore, using bioactive-rich ingredients in the production of novel cookies with enhanced nutritional value is of growing interest [14]. This research aimed to evaluate the nutritional and antioxidant properties of Maya nut flour (MNF) from seeds dried by different methods (sun-dried and hot air at 45 °C and 60 °C), to explore its application in cookies, and to evaluate its nutritional and functional properties. The incorporation of Maya nut in the cookies should improve the nutritional value in terms of protein and fiber, in addition to increasing its antioxidant activity [4,5,6,10].

## 2. Materials and Methods

### 2.1. Plant Material

Two types of seeds were provided by the community of Nuevo Antonio, municipality of Venustiano Carranza, in the state of Chiapas in Mexico: naturally sun-dried seeds that had been stored for six months and fresh (non-dried) seeds collected in March 2021. Leaves, branches, and spoiled seeds were discarded. We husked the seeds and disinfected them with a sodium hypochlorite solution at 200 ppm. In addition to the seeds mentioned above, we purchased commercial MNF from the Ruez catalog, produced by Productos Orgánicos La Selva in Tres Garantias, Quintana Roo, Mexico. Finally, we purchased the wheat flour and other ingredients at a local market in the city of Saltillo, Coahuila, Mexico.

### 2.2. Physical Characteristics of Maya Nut

#### Measurement of Seed Dimensions

We determined the physical measurements of Maya nuts as described by Joshi et al. [15], with some modifications. We used an OHAOUS analytical balance with an accuracy of 0.001 g to determine the weight of 25 seeds. Their main dimensions, length (*L*), width (*W*), and thickness (*T*), were measured using a vernier caliper. The determination of mean diameters (*D_a_*), geometric diameters (*D_g_*) and sphericity (*ψ*) were performed by adopting the equations of Palilo et al. [11]:(1)$$D_{a}=\left(\frac{L+W+T}{3}\right)$$
(2)$$D_{g}={\left(LWT\right)}^{1/3}$$
(3)$$\Psi =\left(\frac{D_{g}}{L}\right)\times 100$$

### 2.3. Drying and Flour Preparation

We used three drying methods to obtain MNF and to compare the effects they demonstrate. The first method was natural sun drying [4]. The seeds were collected and left on rectangular platforms exposed to the sun for five days, then stored in bottles for later use.

For the second and third drying methods, we used an electric food dehydrator (Excalibur 3926TB) at 45 °C for 27 h (second method) and at 60 °C for 24 h (third method). Drying temperatures were selected following reported Maya nut drying methodologies [16,17]. The dried seeds were allowed to cool to 25 °C and stored in bottles until further use.

For flour preparation, we ground the dried seeds in a high-speed multifunction grinder and sieved them on 100 mesh in Cole-Parmer equipment. Finally, the flours were kept in an airtight stand-up pouch and stored in a dry place for further analysis.

The flours obtained were named as follows: naturally sun-dried flour (NF), flour dried at 45 °C (F45), flour dried at 60 °C (F60), and commercial Maya nut flour (CF) (made from roasted seeds at 180 °C for 30 min).

### 2.4. Flour Characterization

#### 2.4.1. Proximal Composition

The nutritional analysis of the MNF was conducted as stated by the Association of Official Analytical Chemists (AOAC) [18]. We used the following methods: fat (AOAC 945.16), ash (AOAC 920.181), crude fiber (AOAC 962.09), and total protein (total nitrogen∗6.25) (AOAC 978.02). Then, we quantified carbohydrates by the difference method. The energy was determined with Equation (4) as reported by Food and Agriculture Organization.
(4)$$Energy\left(Kcal\right)=4\left(g\;protein+g\;carbohydrates\right)+\left(9+g\;fat\right)$$

#### 2.4.2. Mineral Content

X-ray fluorescence spectrometry (XRF) is a non-destructive analytical technique used to obtain elemental information on different types of materials. We placed 1 g of the flours in an energy dispersive X-ray fluorescence spectrometer (ED-XRF) (Epsilon 1, Malvern Panalytical, Madrid, Spain) to determine the mineral content of each flour. The mineral content analysis was performed using the Omnian^®^ software for a period of 20 min per sample.

#### 2.4.3. Techno-Functional Properties of Maya Nut Flours

##### Water Activity

The water activity of all the flours was measured using a water activity meter (AquaLab series 3 Decagon devices) at 24.7 ± 1 °C.

##### Water and Oil Holding Capacity

The functional properties we measured were water absorption capacity (WAC) and oil absorption capacity (OAC). A gravimetric method determined each flour’s capacity to absorb water or oil. We weighed and poured 0.1 g of each flour into a 50 mL Falcon centrifuge tube; after that, we added 10 mL of distilled water or corn oil. The suspensions were vigorously mixed for 5 min using a vortex stirrer and allowed to stand at room temperature (25 °C) for 24 h. We then centrifuged the samples at 3000 rpm for 20 min. The supernatant was then decanted, and the sediment was reweighed. The weight change was expressed as percentage of water or oil absorption based on the weight of the original sample. We calculated this based on the following equation [19]:(5)$$WAC/OAC\left(\%\right)=\left(\frac{w_{2}-w_{1}}{W_{1}}\right)\times 100\%$$
where *WAC*/*OAC* is the water/oil absorption capacity of the flours (%), and *W*_1_ and *W*_2_ are the initial and final weight of the sample (g).

#### 2.4.4. Color

We poured the MNF into Petri dishes until a 10 mm thick layer was formed. A colorimeter (3nh NR110 Precision Colorimeter) measures the *L**, *a** and *b** parameters of the International Commission on Illumination (CIE) of all samples. Corresponding *L** value lightness of color from zero (black) to 100 (white); *a** value (degree of redness [0 to 60] or greenness [0 to 60]); and *b** values (yellowness [0 to 60] or blueness [0 to 60]) were measured for all the samples. Three readings were taken at different locations on the surface of the samples to obtain the mean values of *L**, *a**, and *b**.

#### 2.4.5. Morphological Characteristics

We used a tabletop scanning electron microscope to observe the microstructure of MNF (TM3000, Hitachi, Ltd., Tokyo, Japan). The images were taken at an acceleration voltage of 15 kV and a magnification of 2000.

#### 2.4.6. X-ray Diffraction

The X-ray diffractometer (Panalytical Empyrean, Almelo, The Netherlands) generated a diffractogram with an interval of 2θ angles that ranged from 7 °C to 50 °C at a rate of 4.3°/min and operating at a power of 40 kV/30 mA.

#### 2.4.7. ATR-FTIR Spectroscopy

A PerkinElmer Frontier spectrometer with a universal attenuated total reflection (ATR) determined the main functional groups in the flour and worked in transmittance mode with a resolution of 4 cm^−1^ in the region from 4000 cm^−1^ to 600 cm^−1^ at room temperature.

### 2.5. Total Polyphenol Content and Antioxidant Activity

#### 2.5.1. Extraction

The extraction of total polyphenols and antioxidant components was performed in the dark at room temperature. We weighed 100 mg of flour and mixed it with 5 mL of ethanol/water (1:1 *v*/*v*) solution, following Moo-Huchin et al. [6]. Subsequently, we placed the samples in ultrasound bath equipment (Branson 5510, Marshall Scientific, Hampton, NH, USA) at 25 °C for 10 min. The extracts were then centrifuged at 4500× *g* at 25 °C for 10 min, and the supernatant was recovered and stored in the refrigerator, protected from light, until analysis [20].

#### 2.5.2. Total Phenolic Content

The TPC was estimated using the Folin-Ciocalteu procedure according to the method reported by Goiris et al. [21], with some modifications. A total of 200 μL of the extracts were mixed with 1.5 mL of the Folin-Ciocalteu reagent (previously diluted ten times with distilled water) and allowed to stand at room temperature for 5 min. Then, 1.5 mL of a sodium bicarbonate solution (60 g L^−1^) was added. After a 90 min incubation at room temperature, we measured the absorbance at 750 nm using a spectrometer (Epoch™ microplate, Biotek, Winooski, VT, USA). A standard curve of gallic acid was prepared from 0 ppm to 400 ppm in 50% methanol. Phenolic content was expressed in gallic acid equivalents per g of sample (mg GAE/g).

#### 2.5.3. Evaluation of Antioxidant Activity

The antioxidant activity was evaluated by 2,2-diphenyl-1-picrylhydrazyl (DPPH), 2,2′-Azino-bis(3-ethylbenzothiazoline-6-sulfonic acid) diammonium salt (ABTS+), and ferric reducing antioxidant power (FRAP) methods, described by González et al. [20] with slight modifications. In each test, 5rolox was used as a reference, and the results obtained were expressed as Trolox equivalents per gram of flour.

For the DPPH assay, we prepared a 60 μM solution of DPPH reagent with methanol. Then, 7 μL of each extract was mixed with 193 μL of DPPH solution at the desired absorbance (0.7 ± 0.02) and kept in the dark for 30 min. The absorbance was determined at 517 nm.

To determine antioxidant activity with the ABTS assay, we prepared the ABTS solution at 7 mM with ethanol and mixed it with 2.45 mM K_2_S_2_O_8_ (1:1 *v*/*v*). This ABTS solution was allowed to stand for 16 h in the dark at room temperature (25 °C ± 2), then it was diluted with ethanol to an absorbance of 0.7 ± 0.02 at 734 nm. Next, 5 μL of each sample and 95 μL of ABTS ± solution were placed in a microplate. Solutions were rested for 1 min before reading at 734 nm.

In the case of the FRAP assay, we prepared the FRAP solution by mixing TPTZ (2,4,6-Tripyridyl-s-triazine) at 10 mM, FeCl_3_ (20 mM), and acetate buffer (0.3 M, pH 3.6) in the proportions of 1:1:10 *v*/*v*/*v*. Then, 10 µL of each of the sample solutions was added to the microplate followed by 290 µL of the FRAP solution. Subsequently, we incubated the microplate in the dark at 37 °C for 15 min. The absorbance was measured at 593 nm.

### 2.6. Development of Maya Nut Cookies

The incorporation of MNF in the formulation of cookies was explored. We chose to develop the cookies with NF flour because it showed better techno-functional, nutritional, and functional properties than the F45 and F60. We compared the NF flour to the commercial flour (CF). We replaced 0% (F0), 20% (F20), 40% (F40) or 60% (F60) of wheat flour with NF or CF flour, and we obtained four formulations with substitutions and one control formulation. The ingredients (Table 1) were incorporated for 10 min at medium speed in a mixer (Black + Decker MX900) until a homogeneous mixture was obtained. The dough was rolled out to a thickness of 3 mm, cut into circles with a 5 cm diameter, and baked in a convection oven (HCX Plus 3, San-Son, Naucalpan, Mexico) at 180 °C for 15 min. The cookies were allowed to cool to room temperature and stored in hermetic plastic bags. Then, the nutritional quality and AA of all the formulations were evaluated with the methodologies previously mentioned.

### 2.7. Sensory Evaluation of Maya Nut Cookies

The cookies were made by substituting 20%, 40%, and 60% wheat flour with MNF and compared to a control (100% wheat flour). Sensory acceptance tests of formulations were conducted with 50 semi-trained tasters (aged from 18 to 30 years old) recruited randomly from the staff of the Autonomous University of Coahuila and the Autonomous Agrarian University Antonio Narro. Samples were served in small cups at room temperature. Panelists were asked to show their level of liking for the taste, aroma, texture, color, and appearance attributes for each of the Maya nut cookie formulations. Panelists used a 5-point hedonic scale ranging from 1 (dislike extremely) to 5 (like extremely) with a medium value 3 (neither like nor dislike) to evaluate each formulation, following previously reported methods [14].

### 2.8. Statistical Analysis

All experiments were carried out at least in triplicate and results were reported as the average ± standard deviation (SD). The statistical analysis was performed by one-way analysis of variance (ANOVA) followed by Tukey’s test with a significance level of 5%, using INFOSTAT software version 2018 for Windows (Córdoba, Argentina).

## 3. Results and Discussion

### 3.1. Physical Characteristics of the Maya Nut

The seeds correspond to dicotyledonous seeds, as seen in Figure 1. They are small, have a spherical shape, are covered by a fine yellowish peel, and are dark brown on the outside and green on the inside. The minimum, maximum and mean Maya nut length, width, thickness, weight, geometric mean diameter, and sphericity are shown in Table 2.

*Brosimum alicastrum* grows in areas with average annual temperatures of 18 °C to 27 °C, with annual rainfall of 600 mm (in the Mexican state of Tamaulipas) and up to 4000 mm (in the Mexican states of Chiapas and Tabasco) [22] making it an alternative for utilization and development of new beneficial products.

### 3.2. Proximal Composition of Maya Nut Flours

The water content of the fresh Maya nut was 58.28 ± 4.24%. We obtained all MNF after drying and grinding. The moisture content and chemical composition of the flours are shown in Table 3.

NF’s moisture was higher (9.32%) than F45′s (6.05%) and F60′s (5.92%) moisture. This means that the drying method directly affects this parameter. Low moisture values could result in improved stability for long-term storage with an impact on improving the quality and stability of the product [23].

The ash content of the flours was highest for NF (3.64 ± 0.11 g/100 g), while the lowest was found in F45 (3.08 ± 0.01 g/100 g). The highest protein value was also recorded for NF (6.35 ± 0.44 g/100 g). The lowest protein content was for flour F45 (4.48 ± 0.07 g/100 g). There were no significant differences between crude fat values, which ranged from 1.24 to 1.62 g/100 g across all flours. The highest carbohydrates value was for F45 (84.87 ± 0.11 g/100 g), even higher than CF.

These results differ from those obtained by Subiria-Cueto et al. [4], who reported higher protein content (11.5 ± 0.39%), and lower fat (0.6 ± 0.00%), total carbohydrates (71.2 ± 0.56%) and crude fiber (3.4 ± 0.13%) values than those found in this investigation. On the other hand, Pérez-Pacheco et al. [17] reported lower values in flours from dried seeds at 70 °C for protein (2.02 ± 0.06%) and higher values for crude fat (10.49 ± 1.67%) than the values we obtained. The differences in their proximal composition may be due to the different origins of the seeds and different drying methods employed. To date, few investigations have reported the chemical and nutritional composition of Maya nut, but our results demonstrate the importance of the drying process to which the Maya nut is subjected. Results suggest that natural sun-drying increases the protein concentration.

Traditionally, in some regions of Mexico, MNF has been mixed with other flours, such as corn flour [8]; foods such as tortillas have been prepared and have shown to be a nutritionally enriched product [4]. This occurs because cereals such as corn and wheat are deficient in essential amino acids and in dietary fiber [24], which is improved by the addition of MNF.

In addition, we report the mineral content of Maya nut flour by XRF for the first time in the literature. At least 15 minerals were detected in the samples evaluated, but the most abundant minerals are shown in Table 3. MNF contained mainly potassium (2608.81 ± 0.10 mg/100 g) and calcium (829.08 ± 0.02 mg/100 g), which are two of the most important macrominerals in the daily diet, [25] whereas iron was the third most abundant mineral (60.23 ± 0.10 mg/100 g). Other essential elements found in the samples were zinc, manganese, sulfur, chlorine, and copper. Subiría et al., reported a lower content of potassium, iron, zinc, and sodium, than the mineral’s content we found. This could be due to the different techniques used in the determination of these minerals [4], in addition to the location of the Maya nut, time of harvesting, and sample preparation processes.

### 3.3. Techno-Functional Properties

Water content and Aw are two important factors affecting food safety [23] as values greater than 0.7 allow microbial growth. With that in mind, Table 4 shows that the water activity (Aw) of the MNF was affected by the drying treatment. As the drying temperature increased, the Aw decreased, which led to the naturally sun-dried flour having the highest Aw (0.416 ± 0.01).

The water absorption capacity (WAC) and oil absorption capacity (OAC) of MNF are also shown in Table 4. The observed WAC values ranged from 189.39 ± 10.35% for F45 to 220.32 ± 7.58% for NF flour.

All the MNF we analyzed had higher WAC (189.39% to 220%) than the WAC in whole wheat flour (103.36%), amaranth flour (136.64%), soybean flour (202.26%), lupine flour (155.17%), and corn starch (106.05%) reported by Liu et al. [26]. The ability of flour to absorb water is primarily due to the content of hydrophilic compounds, such as carbohydrates, proteins, and dietary fiber [27].

OAC facilitates emulsification and is another critical attribute in ingredients intended to be incorporated into formulations with high fat content [14]. OAC values of F45 (153.22 ± 3.08) and F60 (157.27 ± 7.94%) were lower than OAC values of NF (173.34 ± 6.46%) and CF (185 ± 4.96%). However, these values were higher than the OAC of wheat flour (146.00% ± 8.94%) recorded by Chandra et al. [14]. Adding components with higher WAC and OAC values to food formulations can contribute to improving certain product’s properties, such as viscosity, stability, and texture [13].

### 3.4. Color

Table 5 shows the results of the color of the MNF. The flour’s color was significantly affected by the drying treatments. The flours obtained from fresh seeds were lighter and had fewer reddish and yellowish pigments than the flour from dried seeds or CF. The values of *L** (lightness) ranged from 82.44 ± 0.27 (in F45) to 58.12 ± 1.08 (in CF). CF’s *a** (redness, 13.30 ± 0.12) and *b** (yellowness, 27.36 ± 0.19) values were significantly higher than the values of all the MNF. Both F45 and F60 showed lower values of redness and yellowness; F45′s redness value was 2.04 ± 0.05, and its yellowness value was 17.60 ± 0.04, whereas F60′s redness value was 3.38 ± 1.06 and yellowness value was 17.46 ± 0.76. A darkening of flours obtained from roasted grains and cereals has been observed previously [28]. The increase in *a** and *b** values in CF flour could be attributed to Maillard reactions caused by chemical reactions between sugars and proteins in the Maya nut during the drying process [29]; for this reason, it is better to use controlled-temperature drying as it does not affect the color parameters. In addition, Maillard reactions generate brown-colored compounds, such as melanoidins; they could also affect the antioxidant properties of foods exposed to high temperatures since it has been observed that toasting increases the antioxidant capacity of foods [30].

### 3.5. Morphological Characteristics

Figure 2 shows the scanning electron micrographs of MNF. The particles observed in all the flours were round, oval, and irregular; these particles correspond to starch granules. These particles were expected to be observed since MNF has been reported to have a 61% starch content [17]. They were unrestrained and exposed, contrary to those observed by Olguin-Maciel et al. [31] who reported starch granules trapped in networks of protein or fibers. We did not observe large agglomerations; the surfaces are smooth and uniform with the presence of some proteins.

The size of the granules was heterogeneous, with an average diameter ranging from 7 to 10 μm. Lindeboom, Chang, and Tyler (2004) [32] categorized the size of starch granules as large (>25 μm), medium (10–25 μm), small (5–10 μm), and very small (<5 μm). According to them, the granules found in this study are small. The size of the granules was similar to those reported in MNF and Maya nut starch by Pech-Cohuo et al. [33] and Pérez-Pacheco et al. [17].

The change in amylose’s structure could have altered and transformed the oval structure of the starch granule into an amorphous shape; alternatively, the whole starch granule may have collapsed [34]. Regardless, the drying temperatures did not seem to affect the structure of the amylose in our MNF. Gels are not visible in Figure 2, which suggests that the drying methods did not cause the starch gelatinization in MFN [35].

### 3.6. X-ray Diffraction Analysis

Figure 3 shows the XRD patterns of the MNF obtained from seeds dried at different temperatures. In plants, starch is stored as a discrete semicrystalline granule and consists of two main components: linear amylose and highly branched amylopectin. According to their X-ray diffraction (XRD) patterns, there are starch polymorphs of type A (mainly in cereal starches), B (found in tubers), and C (A-type and B-type polymorphs) [36]. All MNF exhibited a XRD starch pattern of type C crystals. In MNF, the XRD peaks with the highest intensity for the 2θ angles were 15°, 17°, 18°, and 23°, with minor peaks at 20° and 26°. The drying temperatures did not alter the flour’s crystalline structure, but it did reduce the diffraction intensity as shown by the shape and size of the crystalline peaks [23].

Hay Moo-Huchin et al. [37] mention that the crystallinity is exclusively associated with the packing of the amylopectin double helices, while the amorphous regions are primarily amylose.

### 3.7. ATR-FTIR Spectroscopy

The effect of the different drying methods on the changes in the molecular properties of the obtained flours was determined by a spectroscopic analysis of FTIR bands, and the results can be seen in Figure 4. The signals showed similarity in the position of the bands obtained between the spectra of the flour samples, meaning that the drying methods did not affect the molecular structure of the flours and did not generate different compounds in them. The FTIR spectra show an absorption band at 3260 cm^−1^ corresponding to the stretching vibrations of the hydroxyl groups (–OH) that contributed to the inter and intramolecular interactions of the hydroxyl group OH–, which is a particular characteristic of the starch structure. An intense band at 2920 cm^−1^ is attributable to stretching vibrations of the C–H bonds [12].

Water absorbed by starch is observed at 1620 cm^−1^. The bands at 1400 cm^−1^ and 1300 cm^−1^ correspond to C–H bonds. Starch’s fingerprint peaks are at 1150 cm^−1^ and 1056 cm^−1^; peaks at 1019 cm^−1^ are attributed to the vibrations of the C–O–C glucose bonds, and bands at 928 cm^−1^, 927 cm^−1^, 856 cm^−1^, 758 cm^−1^, and 678 cm^−1^ correspond to the pyranose ring [33].

### 3.8. Total Phenolic Content and Antioxidant Activity

We studied the antioxidant activity (AA) of the MNF because of the importance of looking for new sources of antioxidants for their multiple health benefits as well as their ability to be used as natural preservatives in food [38]. Table 6 shows the results of TPC and antioxidant activity (as determined by DPPH, ABTS, and FRAP methods) of the MNF developed from seeds dried using different methods. The average TPC of the evaluated flours was 12.48 ± 0.81 mg GAE/g. This result is similar to the most recent data recorded in another study for TPC in Maya nut (12.30 ± 10.61 mg GAE/g) [6]. However, it is important to note that the total amount of polyphenols in the flours was significantly affected by the drying method used. Even when significant differences exist, the TPC values are similar between the samples; F60 had the lowest value, with a TPC of 11.61 ± 0.79 mg GAE/g, whereas F45´s TPC increased to 12.78 ± 0.86 mg GAE/g. Nevertheless, CF’s TPC (13.35 ± 0.80 mg GAE/g) was significantly higher than the TPC from all the MNF. This may be because the commercial sample was treated and processed under different conditions than in this analysis and because of its different geographical origin. The Maya nuts used for developing NF, F45, and F60 were collected in the state of Chiapas, and the results of our analysis show a behavior where their TPC is affected by increasing the drying temperature. This highlights the relevance of the geographical origin of plant materials.

The first polyphenolic compounds identified in the Maya nut were epicatechin, hydroxybenzoates, gallic, p-hydroxybenzoic, vanillic acids, hydroxycinnamates, and caffeic and p-coumaric acids. These polyphenolic compounds may contribute to the antioxidant activity of the seeds [5].

In this study, three common tests were performed to measure the AA of Maya nut flours. Significant differences exist between treatments in antioxidant activity (Table 6). The DPPH’s results show a higher AA in F45 (8.49 ± 0.99 μM TEAC/g) than in F60 (2.04 ± 0.47 μM TEAC/g); this is a 4-fold increase. However, ABTS’s results show no significant difference between F60’s AA (68.18 ± 3.66 μM TEAC/g) and F45’s AA (69.66 ± 2.88 μM TEAC/g). Nevertheless, FRAP’s AA results show a 30% increase in F45′s AA with a value of 10.95 ± 0.44 μM TEAC/g. These data show that the MNF had a high TPC, and they exhibited strong free radical scavenging activity and iron-reducing properties. This makes MNF a potential source of natural antioxidants to prevent oxidative damage caused by free radicals [39]. However, the different drying treatments significantly affected the TPC of Maya nuts; drying at 60 °C causes the largest decrease in TPC and AA, which can be seen by the reduction in the ability to remove free radicals from both DPPH and ABTS, as well as the reduction in the iron reducing property (lower FRAP values). Previous research on oven-dried pumpkin flour indicated that higher temperatures (70 °C) increased phenol content [13]; this was also observed in Lemon myrtle leaves [9]. Prolonged exposure to high temperatures during roasting of coffee beans, however, caused the destruction of polyphenolic compounds and the generation of other substances, such as melanoidins that have AA [40].

In the three methods we evaluated, CF has a higher TPC, which leads to a higher AA than the TPC and AA of the other flours. Another cause of this increase in AA could be the generation of other antioxidant compounds during the drying process. Maya nut has emerged recently as a commercial process, and it is common for roasting to be used similarly to the method used for coffee beans. Most of the natural antioxidants available in foods are unstable and sensitive to intense and prolonged heat treatment, which can cause their loss and their biological activity. However, little or no change in the content and activity of these compounds has been observed, though this may vary depending on the food matrix [41]. Furthermore, DPPH and ABTS radical scavenging assay are useful to determine antioxidant capacity, but it is known that the antioxidant capacity detected by ABTS is mainly associated with the content of phenolic and flavonoid compounds, and DPPH mainly reflects the presence of highly pigmented and hydrophilic antioxidants [42]. Roasting can cause changes in the chemical composition and biological activity of the Maya nut seed. It can lose phenolic compounds, while other antioxidant compounds such as melanoidins can be produced. Despite the losses of polyphenolic compounds caused by exposure to intense heat treatments, the generation of these compounds can maintain and even increase the AA [30,40].

### 3.9. Nutritional Characterization of Maya Nut Cookies

Figure 5 shows the results obtained from the nutritional characterization (crude protein, crude fiber, fat, and total carbohydrates) of the Maya nut cookies developed with NF and CF.

The fiber content in cookies made with 60% NF is higher than the fiber content in cookies made with CF. With the substitution of non-conventional flours in the production of functional cookies, the amount of fiber increases and favors the amount of available carbohydrates; this can contribute to a more adequate energy supply in the body and ameliorate consumers’ health [43].

On the other hand, the protein content is directly affected as the flour concentration increases from 0% to 40% in NF cookies (13.5 g/100 g at 40% substitution). For CF cookies, protein content decreases with increasing concentration of CF (5 g/100 g at 60% substitution). Protein content could be affected by the treatment given to the commercial sample since thermal processes can directly affect the protein content; heat treatment can reduce the amount of protein by destroying some amino acids, which modifies the quality of the protein composition of the cookies.

The carbohydrate concentration shows a significant increase as the amount of NF increases. This can be observed in CF cookies, which obtained the highest content (71.9 g/100 g) with 60% substitution; while using NF, the highest value of total carbohydrates occurs with 20% substitution of Maya nut flour (68.4 g/100 g).

Fat content in the cookies substituted with NF showed a constant increase depending on the substitution percentage, so the highest concentration was observed in the 60% substitution. The opposite was seen when CF was substituted in the cookies, so the higher fat concentration occurred in the 20% substitution. Fat content is of substantial relevance in cookie doughs, mainly in the rheological factors; the presence of a greater amount of fats helps to improve handling in elaborating cookies, although little has been published on the subject.

Henrique-Belmiro et al. [44] evaluated cookies that incorporated coffee residues and mentioned that the partial substitution of flour does not cause changes in the dimensions and techno-functional properties of the cookies because of the low levels of residues used in their formulas (3 and 6%). Since we used high substitution levels in our work (0% to 60%), it is reasonable to state that this is the primary reason why we found changes in dimensions and techno-functional properties.

Improving the protein and fiber content in a popular snack, such as cookies, could have a substantial positive effect on its consumers [45]. Other foodstuffs made from Ramon flour include wheat tortillas [4], muffins, and beverages [46]; an increase in nutritional and functional quality, such as increased protein content, dietary fiber content, and antioxidant compounds content, has been registered in these foods. Therefore, Ramon seed flour could have great potential as an enriching ingredient [47].

### 3.10. Antioxidant Activity of Maya Nut Cookies

As discussed earlier, an important component of the Maya nut is its antioxidants. Therefore, to evaluate the stability of this antioxidant capacity when baking cookies, ABTS and DPPH tests were carried out. The results are shown in Figure 6. The behavior of AA in the cookie formulations was similar in the ABTS and DPPH assays. Cookies with 20% of MNF showed the highest AA, but it was lower in cookies with 40% and 60% substitution. This increase in AA was observed in both types of cookies, i.e., cookies containing NF and cookies containing CF. However, between the two types of cookies (NF and CF), the highest increase in AA was in cookies with NF at 20%. Comparing the addition of the two flours, the greatest increase in AA was observed with the addition of NF. However, the substitution of 40% and 60% of wheat flour with NF in the cookies decreased the AA of the product. Meanwhile, the addition of up to 40% CF maintained the AA.

It should be noted that the AA in the cookies can be affected by other ingredients since MNF is not the only source of antioxidants. In this work, wheat flour was substituted with two types of MNF. Although MNF has a higher AA, it was evident that its optimal increase was with the replacement of wheat flour with only 20% of MNF; a higher substitution of MNF did not increase the AA of the cookies; instead, AA was maintained or decreased it, as in the case of the NF. A decrease in AA may also be caused by the antioxidant compounds degrading due to the baking temperatures, since these compounds might be unstable under these conditions [44].

### 3.11. Sensory Evaluation of Maya Nut Cookies

Figure 7 shows the results of the sensory evaluation on a scale ranging from one (extremely dislike) to five (extremely like).

The F20 cookies scored the highest values for aroma (4.15) and flavor (3.92); the panelists expressed a preference for the cookie with Maya nut over the cookie with wheat flour (control). However, the panelists scored wheat cookies with the highest mark (4.2) for the texture attribute. The texture attribute is an essential factor in the quality of foods since it influences the final acceptability of the product [48]. This proves an opportunity to analyze and evaluate the rheological properties of the doughs.

Figure 8 shows the different formulations evaluated and shows a change in color as the concentration of Mayan nut flour increased.

## 4. Conclusions

MNF proved to be an enriching product for food formulations. It can improve the nutritional profile of cookies by increasing the protein content and improve its functional characteristics, such as antioxidant activity, through a 20% substitution of MNF for wheat flour. Few differences were observed in the physicochemical characteristics of the MNF since there was little impact on the structural and molecular characteristics of the starch granules. However, the results of this work demonstrated that the drying methods used here significantly affected the TPC and the AA. Nevertheless, we developed a cookie rich in antioxidants that could ameliorate the damage caused by free radicals in the body and prevent them from causing damage to other cells; this could improve its consumer’s health. In addition, increasing the knowledge about undervalued and little-used products such as Maya nut, along with evaluating the physical, chemical, nutritional, and functional characteristics of ingredients, opens the opportunity to consider them in the development of new food formulations.

## Figures and Tables

**Figure 1 foods-12-01398-f001:**
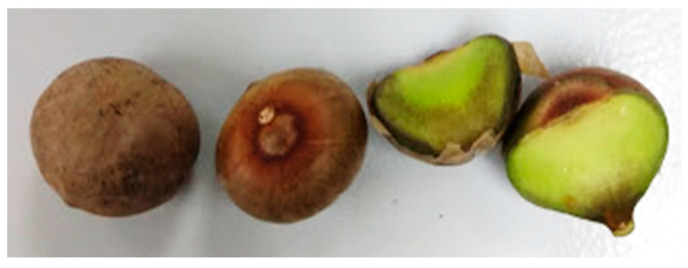
Ramon tree seeds, also called Maya nut.

**Figure 2 foods-12-01398-f002:**
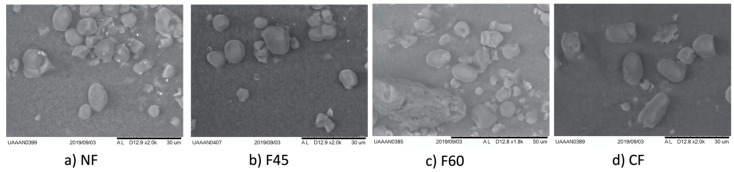
Scanning electron micrographs (magnification at 2000) of (**a**) NF, naturally sun-dried flour, (**b**) F45, flour dried at 45 °C, (**c**) F60, flour dried at 60 °C (**d**) CF, commercial flour.

**Figure 3 foods-12-01398-f003:**
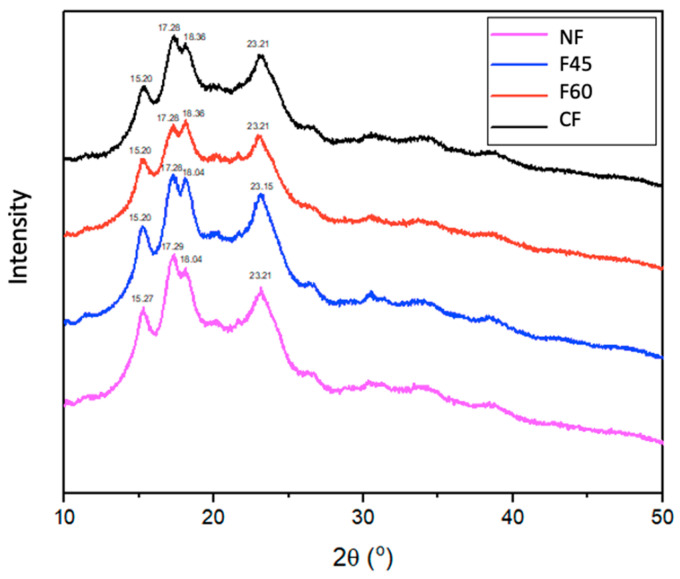
X-ray diffraction (XRD) diagrams of Maya nut flours under different drying treatments. NF—naturally dried flour; F45—flour dried at 45 °C; F60—flour dried at 60 °C; CF—commercial flour.

**Figure 4 foods-12-01398-f004:**
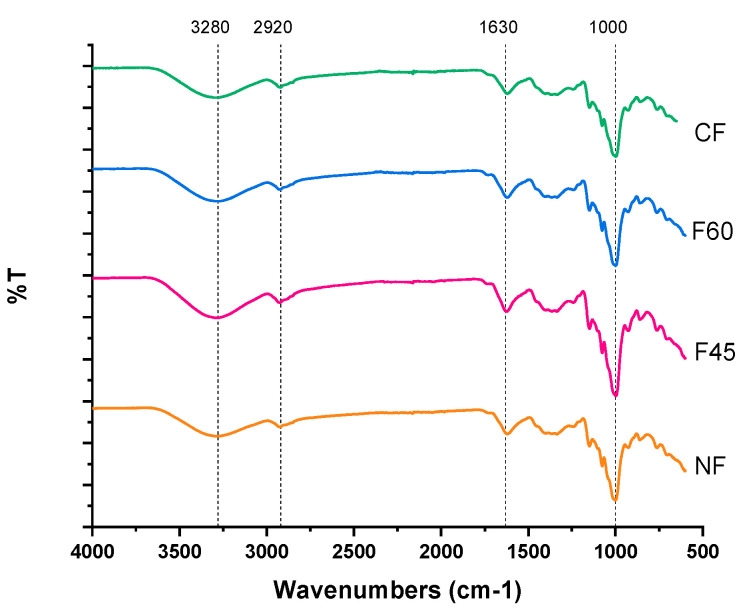
FTIR spectra of Maya nut flour. under different drying treatments. NF-naturally dried flour; F45-flour dried at 45 °C; F60-flour dried at 60 °C; CF-commercial flour.

**Figure 5 foods-12-01398-f005:**
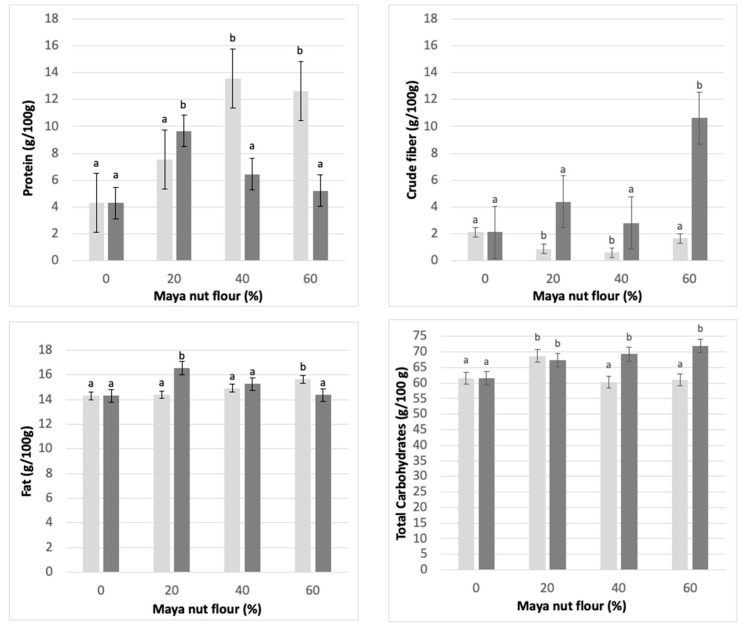
Important nutritional components in Maya nut cookies. ☐ NF (light-colored bar graph), naturally sun-dried flour and ■ CF (dark bar graph), commercial flour. Maya nut flour: 0%, 20%, 40%, and 60%. Different letters indicate significant difference (*p* < 0.05).

**Figure 6 foods-12-01398-f006:**
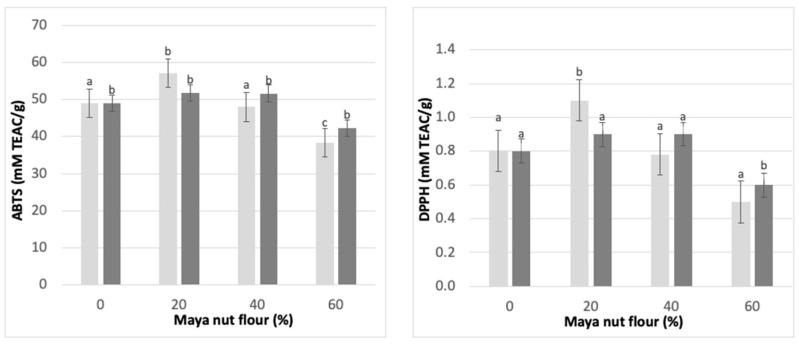
Antioxidant activity in Maya nut cookies by ABTS and DPPH radical. ☐ NF, naturally sun-dried flour and ■ CF, commercial flour. Maya nut flour: 0%, 20%, 40% and 60%. Different letters indicate significant difference (*p* < 0.05).

**Figure 7 foods-12-01398-f007:**
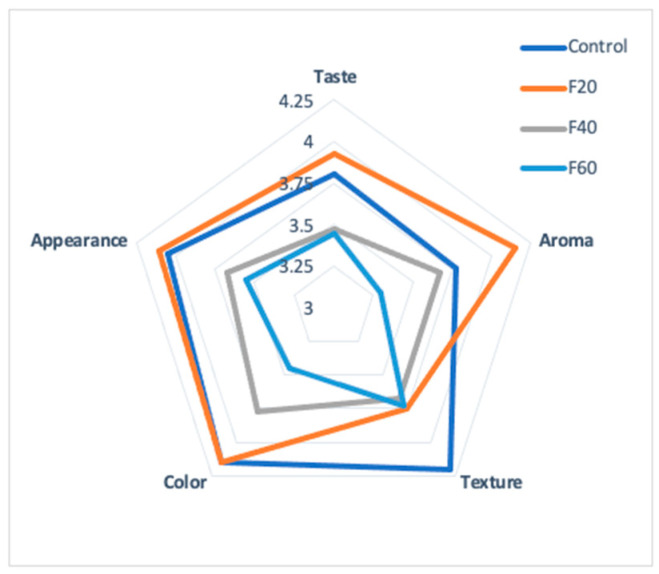
Sensory evaluation of Maya nut cookies. Control (100% wheat flour cookies), F20 (20% naturally sun dried-flour), F40 (40% naturally sun-dried flour) and F60 (60% naturally sun-dried flour).

**Figure 8 foods-12-01398-f008:**
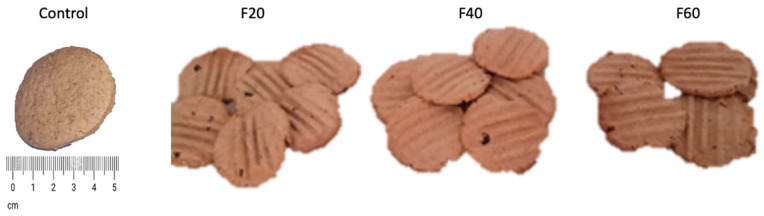
Maya nut cookies. Control (100% wheat flour cookies), F20 (20% naturally sun dried- flour), F40 (40% naturally sun-dried flour) and F60 (60% naturally sun-dried flour). Cookies are 5 cm diameter.

**Table 1 foods-12-01398-t001:** Formulation of two types of Maya nut cookies with commercial flour (CF) and naturally sun-dried flour (NF). F0, 0% MNF substitution; F20, 20% MNF substitution; F40, 40% MNF substitution; F60, 60% MNF substitution.

Ingredients	Quantities (g)			
	F0	F20	F40	F60
Maya nut flour				
- NF	0	20	40	60
- CF				
Wheat flour	100	80	60	40
Milk	45	45	45	45
Butter	30	30	30	30
Honey	11	11	11	11
Egg	6	6	6	6
Walnut	3	3	3	3
Cranberries	3	3	3	3

**Table 2 foods-12-01398-t002:** Physical characteristics of fresh Maya nut.

Parameter	Mean	Minimum	Maximum
Weight (g)	1.71 ± 0.27	1.27	2.20
Length (mm)	18.01 ± 1.05	15.20	20.00
Width (mm)	13.00 ± 0.10	11.01	15.10
Thickness (mm)	17.34 ± 0.81	14.10	19.40
Mean diameter (mm)	16.83 ± 0.82	15.83	17.47
Geometric mean diameter (mm)	16.62 ± 1.20	15.71	17.27
Sphericity (%)	90.83 ± 3.99	86.35	93.88

**Table 3 foods-12-01398-t003:** Nutrimental composition and mineral content of Maya nut flours.

Parameter	NF	F45	F60	CF
Moisture (g/100 g)	9.32 ± 0.19 ^b^	6.05 ± 0.36 ^a^	5.92 ± 0.45 ^a^	6.51 ± 0.29 ^a^
Ashes (g/100 g)	3.64 ± 0.11 ^b^	3.08 ± 0.01 ^a^	3.25 ± 0.29 ^ab^	3.13 ± 0.02 ^a^
Protein (g/100 g)	6.35 ± 0.44 ^c^	4.48 ± 0.07 ^a^	4.93 ± 0.40 ^ab^	5.63 ± 0.36 ^b^
Fat (g/100 g)	1.26 ± 0.11 ^a^	1.52 ± 0.27 ^a^	1.24 ± 0.07 ^a^	1.62 ± 0.73 ^a^
Total Carbohydrates * (g/100 g)	73.83 ± 0.23 ^a^	84.87 ± 0.11 ^c^	84.67 ± 1.06 ^bc^	83.10 ± 0.59 ^b^
Crude Fiber (g/100 g)	6.75 ± 0.29 ^a^	5.30 ± 0.19 ^b^	5.49 ± 0.16 ^b^	4.17 ± 0.46 ^c^
Energy (Kcal)	354.46 ± 0.43 ^a^	371.07 ± 2.78 ^b^	369.52 ± 2.75 ^b^	369.51 ± 4.55 ^b^
Mineral content (mg/100 g)				
Potassium	2608.81 ± 0.10 ^d^	2104.01 ± 0.10 ^a^	2252.31 ± 0.02 ^c^	2185.46 ± 0.03 ^b^
Calcium	829.08 ± 0.02 ^b^	797.35 ± 0.16 ^b^	818.27 ± 0.05 ^ab^	755. 15 ± 0.01 ^a^
Iron	60.23 ± 0.10 ^b^	15.56 ± 0.07 ^a^	18.47 ± 0.10 ^a^	16.18 ± 0.02 ^a^
Zinc	10.09 ± 0.03 ^b^	6.01 ± 0.20 ^a^	7.89 ± 0.35 ^a^	11.67 ± 0.35 ^b^

Mean ± SD. All values are presented on dry weight basis; NF—naturally sun-dried flour; F45—flour dried at 45 °C; F60—flour dried at 60 °C; CF—commercial flour. Different letters indicate significant differences (*p* < 0.05). * Total carbohydrates were determined by the difference method.

**Table 4 foods-12-01398-t004:** Techno-Functional properties of Maya nut flours.

Parameter	NF	F45	F60	CF
WAC (%)	220.32 ± 7.58 ^b^	189.39 ± 10.35 ^a^	204.05 ± 4.02 ^ab^	203.101 ± 6.07 ^ab^
OAC (%)	173.34 ± 6.46 ^b^	153.22 ± 3.082 ^a^	157.27 ± 7.94 ^a^	185.08 ± 4.96 ^b^
Water activity (Aw)	0.416 ± 0.01 ^d^	0.320 ± 0.01 ^b^	0.280 ± 0.01 ^a^	0. 380 ± 0.01 ^c^

Mean ± SD. NF—naturally sun-dried flour; F45—flour dried at 45 °C; F60—flour dried at 60 °C; CF—commercial flour. WAC (water absorption capacity), OAC (oil absorption capacity). Different letters indicate significant difference (*p* < 0.05).

**Table 5 foods-12-01398-t005:** Color parameter of Maya nut flours.

Parameter	NF	F45	F60	CF
*L**	68.09 ± 1.33 ^b^	82.44 ± 0.27 ^d^	75.73 ± 2.20 ^c^	58.12 ± 1.08 ^a^
*a**	6.41 ± 0.38 ^b^	2.04 ± 0.05 ^a^	3.38 ± 1.06 ^a^	13.30 ± 0.12 ^c^
*b**	19.07 ± 0.72 ^b^	17.60 ± 0.04 ^a^	17.46 ± 0.76 ^a^	27.36 ± 0.19 ^c^

Mean ± SD. NF—naturally sun-dried flour; F45—flour dried at 45 °C; F60—flour dried at 60 °C; CF—commercial flour. Different letters indicate significant difference (*p* < 0.05).

**Table 6 foods-12-01398-t006:** Total Phenolic content and antioxidant activity of Maya nut flours.

Parameter	NF	F45	F60	CF
TPC (mg GAE/g)	12.20 ± 0.86 ^ab^	12.78 ± 0.81 ^b^	11.61 ± 0.79 ^a^	13.35 ± 0.80 ^c^
DPPH (μM TEAC/g)	4.78 ± 1.10 ^b^	8.49 ± 0.99 ^c^	2.04 ± 0.47 ^a^	16.87 ± 0.63 ^d^
ABTS (μM TEAC/g)	72.89 ± 1.71 ^b^	69.66 ± 2.88 ^ab^	68.18 ± 3.66 ^a^	68.79 ± 3.46 ^a^
FRAP (μM TEAC/g)	9.76 ± 0.63 ^b^	10.95 ± 0.44 ^c^	7.032 ± 0.08 ^a^	29.84 ± 0.22 ^d^

Mean ± SD. TPC—total phenolic compounds; NF—naturally dried flour; F45—flour dried at 45 °C; F60—flour dried at 60 °C; CF—commercial flour. Different letters indicate significant difference (*p* < 0.05).

## Data Availability

The data presented in this study are available on request from the corresponding author. The data are not publicly available due to privacy.

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
