# Peer review of "Determination of Nutritional and Antioxidant Properties of Maya Nut Flour (Brosimum alicastrum) for Development of Functional Foods"

_foods, 2023, doi:10.3390/foods12071398_

Round 1

Reviewer 1 Report

The current manuscript entitled "Determination of the nutritional and antioxidant properties of the Maya nut flour (Brosimum alicastrum) for the development of functional foods" is about the interesting topic of incorporating bioactives in the food product. However, the structure of the manuscript is not well-written and the current format cannot be considered  unless some major revisions to be done. Moreover, the sensory analysis lacks in this study. I strongly suggest adding the sensory test. My comments are as follows and as in the PDF file:

1. Abstract: the method is not well-presented for both extraction procedure and incorporation into cookies. In abstract, results also should be explained in detail.

2.    Introduction: You should describe a little bit the effects of drying methods on plant materials in the previous study. Moreover, you should write about the previous studies on the incorporation of bioactives from plant materials on cookies and/or other food products. Your hypothesis is not clear.

3.      Materials and methods: you should mention why you used these extraction temperatures. Furthermore, what condition was used for preparation of cookies.

4.      Results and Discussion: The results obtained are not sufficiently elaborated in the present discussion especially for Nutritional and antioxidant characterization of Maya nut cookies.

Author Response

Comment Reviewer

  1. Abstract: the method is not well-presented for both extraction procedure and incorporation into cookies. In abstract, results also should be explained in detail.

Author´s Comments

The reviewer's suggestion was accepted and addressed

Comment Reviewer

  1. Introduction: You should describe a little bit the effects of drying methods on plant materials in the previous study. Moreover, you should write about the previous studies on the incorporation of bioactives from plant materials on cookies and/or other food products. Your hypothesis is not clear.

Author´s Comments

The reviewer's suggestion was accepted and addressed. Relevant information was added to enrich the text, and to be easily understood.

Comment Reviewer

  1. Materials and methods: you should mention why you used these extraction temperatures. Furthermore, what condition was used for preparation of cookies.

Author´s Comments

Information was added in the text so that the methodology and the development of the work are understandable.

Comment Reviewer

  1. Results and Discussion: The results obtained are not sufficiently elaborated in the present discussion especially for Nutritional and antioxidant characterization of Maya nut cookies.

Author´s Comments

The reviewer's suggestion was accepted and addressed. Relevant information was added to enrich this part of the text.

All comments in the pdf were made.

Sensory test was added

Reviewer 2 Report

The work is focused on the study of the nutritional and antioxidant potential of Maya nut flour in order to be used on food enrichment. To this purpose, the drying process of the flour by using a natural sun drying and different drying temperatures have been evaluated. The elemental and chemical composition, the colour, the water and oil absorption capacities, among other properties, were analysed. Finally, two of the analysed flours were used in different proportions to be incorporated into the cookie formulation and the nutritional profile and the antioxidant activity were compared. It is an interesting work in the functional food field that could be helpful to expand knowledge, but it should be deeply reviewed as, frequently, the connection between the text and the results showed is lost and, also, there are several format errors. Hence, the following major revisions should be made:

Introduction:

The introduction is generally well described, but information about the availability and uses of enriched cookies in the market as functional foods is missed.

Materials and Methods:

-        Line 70-71: Specify how long the seeds were sun-dried and the store temperature.

-        Line 72: Specify the concentration of the sodium hypochlorite solution.

-        Check the equations: format of number in equations must be uniform. Also check the numbering, equation (3) is repeated.

-        Line 77: Remove the subsection if there is just one.

-        Line 119: Define the time used for mixing the suspensions.

-        Line 132: Check the spell of “tree”.

-        Line 145: “In” is repeated twice, check.

-        Line 159: Check the reference 2.10.1., should be 2.11.1.

-        Line 175 and 181: adjust properly the numeral for the absorbance value and its error.

-        Line 191: It is said that NF was used for the enrichment of the cookies, but in the results and discussion section, it is also showed the results obtained from the CF sample. Please include it here also.

Results and Discussion:

-        Check the number of the tables, table 2 is repeated. Check and correct also the references in the text.

-        Line 214: In table 2, define the units for “Mean diameter”.

-        Line 218: Why these conditions make the sample interesting for obtaining beneficial products? It is said that they could be an interesting alternative to this purpose, but, alternative to what?

-        Line 223: In table 2 make the units uniform for all the components to percentage. Correct also in the text. How was de energy value determined? Please include it in the materials and methods section.

-        Line 229: Translate “para” to English.

-        Line 249-251: A little review about the total and essential amino acids in Maya nut flour is missed here, in comparison with those presented in corn and wheat cereals.  

-        Line 255-257: Make clearer that these results are from NF samples.

-        Line 259-260: The differences found in the mineral content in comparison with the author Subiría et al. due to the analytical procedure should be more clearly described. Give the reasons.

-        Line 264-265: Explain how water and Aw affect to food safety.

-        Line 275: Use the same code to refer to WAC and OAC. Use these or OHC and WHC, but not exchange. The same in the materials and methods section.

-        Line 276: Correct “&” spelled.

-        Line 282: The statistical analysis for water activity (Aw) is missed. Please review and place the proper number of numerals in the average and the error for the determined parameters.

-        Line 290-292: It is said that flours obtained from unroasted seeds, that is NF, are lighter and lower red and yellow, but in Table 4, the parameters indicate the opposite: lower value for L*, and higher values of a* and b* of NF in comparison with the F45 and F60 flours.

-        Line 294 and 296: Review the values “13.40” and “17.70”, it is not the same than specified in the table below. In line 296, specify what values are for a* and b* parameters.

-        Line 299-304: If during Maillard reactions compounds with high antioxidant activity are generated, why is better to use moderate temperatures under which they are not produced? Why the browning of the flours after these reactions could affect its quality?

-        Line 348: Translate “y” to English.

-        Line 358-387: The occurrence of Maillard reactions is said as a possible cause of the increase in the antioxidant activity of the CF samples. However, in the F60 sample, also produced after a roasting process, the TPC and the antioxidant activity determined by the three different methods, DPPH, ABTS, FRAP, are the lowest in comparison with the other samples. Please give a possible reason.  

-        Line 389: Explain why so big differences were found between ABTS and the rest of the analysis for antioxidant activity and also explain why with ABTS differences among samples were not identified but they were with the other assays.

-        Line 422-425: The sentence seems to be not complete, review.

-        Check the number of the figures: figure 5 is missed. Check also the references in the text.

-        Line 429-431: Rephrase for a better understanding.

-        Line 436: Please include the statistical analysis in the comparison of NF and CF nutritional components (figure 6). Also, re-scale the total carbohydrates figure to fit properly the error bars.

-        Line 438-442: Rephrase. It is not all correct: for NF the protein content increased by increasing the maya nut flour proportion but up to 40%, it remains stable if still increases up to 60%. For CF protein content does not decrease with increasing concentration in all conditions: it increases from 0% to 20% and then, it starts to decrease if maya nut flour increases more. Also, find a reference to support the statement given about protein being affected by the thermal procedure.

-        Line 448-451: Review. It is not correct. For NF sample, the fat proportion seems to be the same for 0 and 20% samples, and similar for 40 and 60%, support with the statistical analysis. Although it was significant different, the samples are very similar in the fat content, between 14 and 16 %.

-        Line 462: Check the numbering, subsection 2.9 is repeated.

-        Line 467: Add the statistical analysis in the figure.

-        Line 470-473: Review, the statement does not agree with the results showed in the graph.

-        Line 429: delete the last reference style.

Conclusions: Some of the most important results obtained in the work should be stated here to support the conclusions reached.

Author Response

Comment Reviewer

Introduction: The introduction is generally well described, but information about the availability and uses of enriched cookies in the market as functional foods is missed.

Author´s Comments

the information was improved and enriched for more clarity on the subject.

Comment Reviewer

Materials and Methods:

  • Line 70-71: Specify how long the seeds were sun-dried and the store temperature.

Author´s Comments

Is done

  • Line 72: Specify the concentration of the sodium hypochlorite solution.

Author´s Comments

Is done… 200 ppm

  • Check the equations: format of number in equations must be uniform. Also check the numbering, equation (3) is repeated.

Author´s Comments

Is done

  • Line 77: Remove the subsection if there is just one.

Author´s Comments

Is done

  • Line 119: Define the time used for mixing the suspensions.

Author´s Comments

Is done, 5 minutes

  • Line 132: Check the spell of “tree”.

Author´s Comments

Is done, three.

  • Line 145: “In” is repeated twice, check.

Author´s Comments

Is done

  • Line 159: Check the reference 2.10.1., should be 2.11.1.

Author´s Comments

The information was rewritten

  • Line 175 and 181: adjust properly the numeral for the absorbance value and its error.

Author´s Comments

Is done

  • Line 191: It is said that NF was used for the enrichment of the cookies, but in the results and discussion section, it is also showed the results obtained from the CF sample. Please include it here also

Author´s Comments

Was included

Comment Reviewer

Results and Discussion:

  • Check the number of the tables, table 2 is repeated. Check and correct also the references in the text.

Author´s Comments

Is done

  • Line 214: In table 2, define the units for “Mean diameter”.

Author´s Comments

Is done

  • Line 218: Why these conditions make the sample interesting for obtaining beneficial products? It is said that they could be an interesting alternative to this purpose, but, alternative to what.

Author´s Comments

Is done

  • Line 223: In table 2 make the units uniform for all the components to percentage. Correct also in the text. How was de energy value determined? Please include it in the materials and methods section.

Author´s Comments

Is done

  • Line 229: Translate “para” to English.

Author´s Comments

Is done

  • Line 249-251: A little review about the total and essential amino acids in Maya nut flour is missed here, in comparison with those presented in corn and wheat cereals.  

Author´s Comments

Is done

  • Line 255-257: Make clearer that these results are from NF samples.

Author´s Comments

Is done

  • Line 259-260: The differences found in the mineral content in comparison with the author Subiría et al. due to the analytical procedure should be more clearly described. Give the reasons.

Author´s Comments

Is done

  • Line 264-265: Explain how water and Aw affect to food safety.

Author´s Comments

Is done

  • Line 275: Use the same code to refer to WAC and OAC. Use these or OHC and WHC, but not exchange. The same in the materials and methods section.

Author´s Comments

Is done

  • Line 276: Correct “&” spelled.

Author´s Comments

Is done

  • Line 282: The statistical analysis for water activity (Aw) is missed. Please review and place the proper number of numerals in the average and the error for the determined parameters.

Author´s Comments

Is done

  • Line 290-292: It is said that flours obtained from unroasted seeds, that is NF, are lighter and lower red and yellow, but in Table 4, the parameters indicate the opposite: lower value for L*, and higher values of a* and b* of NF in comparison with the F45 and F60 flours.

Author´s Comments

The information was rewritten

  • Line 294 and 296: Review the values “13.40” and “17.70”, it is not the same than specified in the table below. In line 296, specify what values are for a* and b* parameters.

Author´s Comments

The information was rewritten

  • Line 299-304: If during Maillard reactions compounds with high antioxidant activity are generated, why is better to use moderate temperatures under which they are not produced? Why the browning of the flours after these reactions could affect its quality?

Author´s Comments

The information was rewritten

  • Line 348: Translate “y” to English.

Author´s Comments

Is done

  • Line 358-387: The occurrence of Maillard reactions is said as a possible cause of the increase in the antioxidant activity of the CF samples. However, in the F60 sample, also produced after a roasting process, the TPC and the antioxidant activity determined by the three different methods, DPPH, ABTS, FRAP, are the lowest in comparison with the other samples. Please give a possible reason.  

Author´s Comments

It is because these samples were dehydrated and the CF was roasted (company or supplier conditions).

  • Line 389: Explain why so big differences were found between ABTS and the rest of the analysis for antioxidant activity and also explain why with ABTS differences among samples were not identified but they were with the other assays.

Author´s Comments

Is done

  • Line 422-425: The sentence seems to be not complete, review.

Author´s Comments

The information was rewritten

  • Check the number of the figures: figure 5 is missed. Check also the references in the text.

Author´s Comments

Is done

  • Line 429-431: Rephrase for a better understanding.

Author´s Comments

Is done

  • Line 436: Please include the statistical analysis in the comparison of NF and CF nutritional components (figure 6). Also, re-scale the total carbohydrates figure to fit properly the error bars.

Author´s Comments

Is done

  • Line 438-442: Rephrase. It is not all correct: for NF the protein content increased by increasing the maya nut flour proportion but up to 40%, it remains stable if still increases up to 60%. For CF protein content does not decrease with increasing concentration in all conditions: it increases from 0% to 20% and then, it starts to decrease if maya nut flour increases more. Also, find a reference to support the statement given about protein being affected by the thermal procedure.

Author´s Comments

Is done

  • Line 448-451: Review. It is not correct. For NF sample, the fat proportion seems to be the same for 0 and 20% samples, and similar for 40 and 60%, support with the statistical analysis. Although it was significant different, the samples are very similar in the fat content, between 14 and 16 %.

Author´s Comments

Is done

  • Line 462: Check the numbering, subsection 2.9 is repeated.

Author´s Comments

Is done

  • Line 467: Add the statistical analysis in the figure.

Author´s Comments

Is done

  • Line 470-473: Review, the statement does not agree with the results showed in the graph.

Author´s Comments

The information was rewritten

  • Line 429: delete the last reference style.

Author´s Comments

Is done

Comment Reviewer

Conclusions: Some of the most important results obtained in the work should be stated here to support the conclusions reached.

Author´s Comments

Is done

Reviewer 3 Report

Manuscript present 4 type of flours based on Maya nut, but no control (blank) sample is described. Same problem, absence of the control (blank) at the formulation of Maya cookies. I recommend to define a control sample to compare with the other samples. 

In table 2 appear the Energy(kcal)  but in methodology is not described the calculation method. 

Line 229 please replace "para" with "for".

From line 247 to 251 manuscript presents Maya nut flour mixed with corn flour, but the study refer only to wheat flour. 

Author Response

Comments and Suggestions for Authors

Manuscript present 4 type of flours based on Maya nut, but no control (blank) sample is described. Same problem, absence of the control (blank) at the formulation of Maya cookies. I recommend to define a control sample to compare with the other samples. 

Author´s Comments

The control was considered the 100% wheat flour cookie or the 0% MNF biscuit.

Comment Reviewer

In table 2 appear the Energy(kcal)  but in methodology is not described the calculation method. 

Author´s Comments

Comment Reviewer

Line 229 please replace "para" with "for".

Author´s Comments

Is done

Comment Reviewer

From line 247 to 251 manuscript presents Maya nut flour mixed with corn flour, but the study refer only to wheat flour. 

Author´s Comments

The reviewer's comment was addressed.

Reviewer 4 Report

The presented article determines the nutritional properties and the antioxidant activity of the Maya nut flours for the production of cookies, including different drying methods. While the article offers an interesting contribution to the field, some minor alterations could improve its academic quality.

Specific comments:

Pages 1 and 2, To enhance the introduction section, additional relevant references could be included to provide a more comprehensive overview of the topic. Furthermore, an explanation should be provided to justify why Maya nut flour cookies can be classified as a functional food.

Page 2, lines 63-66, the article objective is unclear, missing the Maya nut flour in the explanation.

Pages 2-5, In the material and methods section, it is recommended to specify the type and producer of the equipment used to increase the reproducibility of the experiments. Page 5, Table 1, add the producers for every ingredient used in the cookie formulation,

Page 6, Table 2, add the initial water content of the fresh Maya nut

Page 8, Table 4, calculate ΔE improve the data analysis.

In the results section, the addition of photos of the final product would be a valuable addition to the article.

Page 13, line 479, remove highlighting.

To enhance the academic quality of the paper, it is recommended that a thorough check of the English language.

Overall, these minor modifications could improve the academic quality of the article and provide a more comprehensive analysis of the nutritional properties and antioxidant activity of Maya nut flours for the production of cookies.

Author Response

Comment Reviewer

Pages 1 and 2, To enhance the introduction section, additional relevant references could be included to provide a more comprehensive overview of the topic. Furthermore, an explanation should be provided to justify why Maya nut flour cookies can be classified as a functional food.

Author´s Comments

The reviewer's suggestion was accepted and addressed. Relevant information was added to enrich the introduction and with this it is intended that the background and generalities of the work be easily understood.

Comment Reviewer

Page 2, lines 63-66, the article objective is unclear, missing the Maya nut flour in the explanation.

Author´s Comments

The reviewer's recommendation was met. Line 140-146. Reworded the objective to be clear.

Comment Reviewer

Pages 2-5, In the material and methods section, it is recommended to specify the type and producer of the equipment used to increase the reproducibility of the experiments. Page 5, Table 1, add the producers for every ingredient used in the cookie formulation,

Author´s Comments

Suggested information was added to the document.

Comment Reviewer

Page 6, Table 2, add the initial water content of the fresh Maya nut

Author´s Comments

Suggested information was added to the document.

Comment Reviewer

Page 8, Table 4, calculate ΔE improve the data analysis.

Author´s Comments

The reviewer's suggestion is appreciated, but this type of analysis is not contemplated within the scope of the research. This evaluation will be considered for subsequent publications.

Comment Reviewer

In the results section, the addition of photos of the final product would be a valuable addition to the article.

Author´s Comments

Is done

Comment Reviewer

Page 13, line 479, remove highlighting.

Author´s Comments

The highlighted text was removed from the document.

Comment Reviewer

To enhance the academic quality of the paper, it is recommended that a thorough check of the English language.

Author´s Comments

The language of the document was reviewed and corrected.

Reviewer 5 Report

In this manuscript, the authors reported the nutritional properties and the antioxidant activity of the Maya nut flours for the production of functional foods. The topic is interesting. However, there are several issues needed to be addressed.

Q1 : Line 18, ' the different drying method: natural (NF), dehydrater at 45°C (F45) and 60°C (F60). ' Please check your grammar and spelling again carefully.  

Q2 : Line 63, the objective of this research needs to be rewritten. It is difficult to catch the main idea. 

Q3: Line 132, check the writing throughout the manuscript. e.g. change 'Tree' to 'Three' .

Q4: Line 406, please include the scientific reasoning details for this observation. Or do you have some appropriate references to support this statement?

Q5: Line 410, please check your grammar and spelling again carefully. 

Q6: Line 453, do you have more details on the rheological factors to support this statement? 

Q7: Line 458, please check your grammar and spelling again carefully. You should modify the sentences. It is difficult to understand. 

Author Response

Comment Reviewer

Q1 : Line 18, ' the different drying method: natural (NF), dehydrater at 45°C (F45) and 60°C (F60). ' Please check your grammar and spelling again carefully.  

Author´s Comments

The reviewer's comment was addressed. Corrected the spelling and grammar of the highlighted text.

Comment Reviewer

Q2 : Line 63, the objective of this research needs to be rewritten. It is difficult to catch the main idea. 

Author´s Comments

The reviewer's suggestion was accepted and addressed. Reworded the objective to be clear.

Comment Reviewer

Q3: Line 132, check the writing throughout the manuscript. e.g. change 'Tree' to 'Three' .

Author´s Comments

The reviewer's comment was addressed.

Comment Reviewer

Q4: Line 406, please include the scientific reasoning details for this observation. Or do you have some appropriate references to support this statement?

Author´s Comments

The text was redrafted to make the discussion more understandable.

Comment Reviewer

Q5: Line 410, please check your grammar and spelling again carefully. 

Author´s Comments

The reviewer's comment was addressed.

Comment Reviewer

Q6: Line 453, do you have more details on the rheological factors to support this statement? 

Author´s Comments

Is done

Comment Reviewer

Q7: Line 458, please check your grammar and spelling again carefully. You should modify the sentences. It is difficult to understand. 

Author´s Comments

The reviewer's comment was addressed.

Round 2

Reviewer 1 Report

The authors have made enough improvements. So, the current version of the manuscript can be considered for the publication.

Reviewer 2 Report

The authours have considered the reviews properly, so I recommend the article to be published in the present form.

Reviewer 5 Report

Accept in present form.